# Telemedicine Treatment and Care for Patients with Intellectual Disability

**DOI:** 10.3390/ijerph18041746

**Published:** 2021-02-11

**Authors:** Krzysztof Krysta, Monika Romańczyk, Albert Diefenbacher, Marek Krzystanek

**Affiliations:** 1Department and Clinic of Psychiatric Rehabilitation, Department of Psychiatry and Psychotherapy, Faculty of Medical Sciences, Medical University of Silesia in Katowice, Ziołowa 45/47, 40-635 Katowice, Poland; romanczykmonika@wp.pl (M.R.); krzystanekmarek@gmail.com (M.K.); 2Department Psychiatry and Psychotherapy (CBF), Charité University Medicine Berlin, Herzbergstraße 79, 10365 Berlin, Germany; albert.diefenbacher@charite.de

**Keywords:** intellectual disability, telemedicine, telepsychiatry, e-health, mental disorders

## Abstract

In recent years, telemedicine has been developing very dynamically. The development of new technologies allows their use in the treatment of dermatological, cardiological, endocrine and other diseases. However, there are few reports on the use of digital technologies in the mental health care of people with intellectual disabilities. Intellectual disability is a disease that affects a large number of people. Patients suffering from intellectual disability encounter barriers that make it difficult for them to fully use telemedicine, however, these barriers can be overcome with appropriate support and adaptation. A review of the literature on telemedicine solutions in the care of people with intellectual disabilities indicates that the applications support the communication of these people with the doctor, enable simple behavioral interventions, stimulate cooperation in treatment, provide simple medical education as well as record medical data for the doctor. The authors present the potential risks related to the use of telemedicine solutions for people with intellectual disabilities as well as the project of creating a new, multi-module telemedicine system.

## 1. Introduction

Intellectual disability is a mental disorder that affects a significant number of people. In England, for example, it affects 1 million adults, which is approximately 2% of the country’s population [1]. People with intellectual disabilities experience cognitive and adaptive deficits throughout their lives, which significantly translate into everyday life and make it difficult to obtain education, career development and social contacts [2]. They may also experience difficulties in obtaining adequate medical care.

The purpose of using digital technologies in psychiatry is to help patients to achieve well-being as well as to assess, treat and monitor mental illness. In people with intellectual disabilities, it is important to ensure that they are efficient and safe in using digital technologies [3].

The Internet and digital devices are so widespread that, according to data from 2016, 90% of adults used the Internet every day, while 7 out of 10 adults had a smartphone. Thanks to such widespread distribution, the Internet can be used not only for communication or obtaining information, but also for receiving medical advice via electronic devices [4]. However, among the population of people with intellectual disabilities, the percentage of people using the Internet is much smaller. It is estimated that 25–50% of people with intellectual disabilities use the Internet [5,6]. Likewise, fewer of these patients have a mobile phone, and if they do, they mainly use its basic functions [7]. This barrier limits the possibilities of reaching people with intellectual disabilities through digital technology [8]. It also indicates the need to create applications adapted to the needs and the possibility of using them by people with intellectual disabilities.

There are several barriers that people with intellectual disabilities may encounter and that may discourage them from using modern technologies. First of all, these people show linguistic and cognitive limitations that may make it difficult for them to use text-rich applications. People with intellectual disabilities will need more time to learn new skills necessary to use new technologies. Moreover, they can be discouraged by failed experiences. There are also physical obstacles that may make it difficult for patients to use, e.g., a computer keyboard, and sensory barriers that make it difficult to view information on small screens. Nevertheless, it has been shown that people with mild to moderate intellectual disability can easily learn using the basic options of electronic devices [9,10]. 

People with a higher degree of intellectual disability, with appropriate support from a caregiver, can also use telemedicine applications, although to a limited extent [11]. It is worth paying attention to the fact that people with intellectual disabilities are often unable to work. For this reason, the monetary aspect may be another obstacle. Electronic devices generate costs in the form of the purchase of equipment, payment for internet access, servicing, etc. People with intellectual disabilities are, however, interested in using modern technologies, they see it primarily as an opportunity to expand the social circle and treat it as a way to develop their own interests [12], as well as to have positive and enjoyable leisure experiences [13]. These solutions can be a supportive tool in schools, e.g., to teach logical-mathematical concepts [14]. VR technology can also be used to improve patients’ physical fitness [15].

In overcoming the barriers hindering the use of modern technologies, it is helpful to design telemedicine applications and electronic devices in such a way that they are accessible to every user without the need to adapt them [16]. The development of new technologies should lead to the creation of an intelligent interface that would make it possible for more people to take advantage of the VR applications, including the users who have limitations caused by ageing or degenerative neurological conditions [17].

The aim of the authors was to review the existing research on the possibilities of using telemedicine solutions in the care for people with intellectual disabilities, indicating the possibilities, limitations, and potential risks associated with it, and to present the idea of our own solution for the telemedical care system for these people. 

## 2. Methods

This review was focused on the adoption on telemedicine tools in the treatment of patients with intellectual disabilities. In order to achieve this result, the following databases PubMed, Web of Science and Google Scholar were searched (effective date 20 December 2020). The search was performed according to the PICO framework (P—patient, problem or population, I—intervention, C—comparison, control or comparator, O—outcomes). During our search, we used the following terms: intellectual disability (Title/Abstract), telemedicine, telecare (Title/Abstract), traditional consultations (Title/Abstract), psychical, physical activity improvement (Title/Abstract).

## 3. Telemedicine Applications for People with Intellectual Disabilities

The search provided with the use of above terms resulted in finding 16 manuscripts; however, after limiting the selection only to original papers fulfilling the criteria of good quality research studies, the papers listed in the table (Table 1) were included in the analysis: 

The study by Salgado et al. (2018) is interesting in the context of designing telemedicine applications for the care of people with intellectual disabilities. They checked what options should be offered by a drug management application adapted to young adults with intellectual disabilities. The drug management app feature survey was conducted by searching iTunes and the App Store in February 2016 using the following terms: “adherence”, “drugs”, “drug management”, “drug list”, and “drug reminder”. After identifying the functions in the downloaded applications, a list of 42 functions was finally compiled, grouped into 4 modules (drug list, drug reminder, drug administration record and additional functions), which were included in the questionnaire prepared for evaluation by experts. In total, 52 developmental disability experts were invited to evaluate the questionnaire, including people with developmental disabilities, caregivers and professionals. They were invited to participate in the 3-round Delphi technique. The aim was to reach a consensus on the characteristics that are preferred and appropriate for promoting independence in the drug management process among people with developmental disabilities. Consensus for the first, second and third rounds was defined as ≥90%, ≥80%, and ≥75% agreement, respectively. At the end of the third round, consensus had been reached for 60% (12/20) of the drug list items, 100% (3/3) of the drug reminder module, 67% (2/3) of the drug administration log module, and 63% (10/16) in the extra functions module. The additional functions selected by experts included: providing information on drugs to family members and caregivers, automatic sending of information to the pharmacy about the expiry of drugs and signaling the need to supplement them. Of the 42 different characteristics assessed, 64% (27/42) reached a consensus to include them in a future drug management application specifically tailored for people with intellectual disabilities [18].

The results of the study by Ptomey et al. (2017), who assessed the feasibility of an intervention aimed at increasing physical activity in a group of mentally disabled adolescents via teleconference, are also interesting. The subjects participated in 30 min sessions using electronic devices three times a week; 31 participants were registered and 29 completed the 12-week classes. Participants were present in 77.2 ± 20.8% of scheduled sessions; mean session duration was 26.7 ± 2.8 min, of which 11.8 ± 4.8 min at moderate or high intensity. Based on the study group, the use of telemedicine among people with intellectual disabilities may be a good solution aimed at promoting health in this group [19].

In one of the few clinical studies done by Taber-Doughty et al. (2010), the effectiveness of traditional care with tele-visits was compared in 4 patients with intellectual disability. Traditional care and tele-consult were alternated for 6 weeks. The subjects were to perform specific tasks such as addressing the envelope, cooking pudding, making tea. The tele-consult was provided using the Voice over Internet Protocol (VoIP), which provided audio communication over the Internet. It has been shown that the use of tele-visits, despite the fact that it extended the time of task completion, increased the independence of the respondents. The authors emphasize, however, that due to the very small size of the studied group, the results of this analysis require confirmation in further studies [20].

Electronic alarm systems have been also designed and used to monitor behavior and activity, for example by alerting the caregiver that an intellectually disabled person is leaving the house. In addition, teleconsultations with a specialist were used, which is especially useful for people with intellectual and developmental disabilities, where there may be very few specialists who can help them, and the visits, if they were to take place directly, would involve walking a long distance [21]. Application for people with intellectual disabilities MY HEALTH GUIDE was created to facilitate the transfer of information and its understanding. The MY CHOICE PAD application, containing several thousand characters and symbols, aims to support people with intellectual disabilities in communication [3].

## 4. Telemedicine among People with Intellectual Disabilities during the COVID-19 Pandemic

The current pandemic situation has, to some extent, forced the public to use healthcare, education and other services through electronic devices. For people with disabilities still some barriers need to be challenged in order to make it easier for them to adopt to this new situation. They are, for example, barriers in infrastructure, access to services, operational challenges, regulations, communication and legislation [22]. For some countries, the fast implementation of telemedicine tools may be complicated by geographic disparities. For example, the US is the frontrunner in the access to tele-health, and on the other hand Bangladesh or Caribbean countries are in the entry stage of its adoption. Other countries like China, Spain, Italy, UK, Australia, Latin American countries are on varying levels of access to this type of support, with several problems like regional differences, technological and legal aspects still to be solved [23].

A survey by Jeste et al. (2020) collected 669 responses from the US and 149 from outside the US. Respondents were parents/caregivers of people with intellectual disabilities. Unfortunately, as many as 74% of parents reported that their child lost access to at least one therapy or educational service, and 36% of respondents lost access to a doctor during the pandemic. Only 56% of people confirmed that their child received continuation of at least some services through electronic devices. The epidemic situation undoubtedly influenced access to health care and education of people with intellectual disabilities and others. However, remote healthcare service, if properly implemented, may be useful not only due to ongoing restrictions, but also after the end of the pandemic [24].

Another study by Zaagsma et al. analyzed the use of online support by people with intellectual disabilities during the COVID-19 pandemic. The COVID-19 outbreak and related containment measures have been shown to have had a strong impact on the use of online support, in particular, there was an increase in the number of unscheduled visits made via the internet [25]. 

## 5. Experiences of the Berlin Treatment Center for Health in Patients with Developmental Disabilities during the COVID-19 Pandemic

During the first lockdown period beginning at the end of March 2020, patients’ and carers’ visits to the outpatient clinic of the center were stopped, and telephone calls were used on a regular basis to maintain the contact. This was very well accepted by patients and carers. In the beginning of June 2020, the possibility of video consultations was provided, however they were performed quite rarely, and telephone calls have been still the preferred method of communication, especially because medical home visits had to be stopped nearly completely. It seems to be the case, that especially carers in nursing homes prefer telephone calls to videoconferencing, as they can be managed more flexibly. In case patients and carers use both phone or video connections, direct communication with the patients, mostly with those with mild or moderate developmental disability is possible, when they are able to use smartphones. Visits to the outpatient clinic are still necessary in case, e.g., of taking blood samples. Here, as well as in the hospital, the vast majority of patients tolerate the use of face masks (mouth-nose protection) quite well. It is of interest that the possibility of using smartphones via voice control seems to lead to an increasing use of such devices in this population, in this way promoting participation in peer groups and society.

## 6. The Use of Telemedicine among People with Intellectual Disabilities in Different Health Conditions

People with intellectual disability are often characterized by high rates of mental illnesses [26]. Digital interventions can bring many benefits to such people in the form of improvement in education or the possibility of vocational development. Preliminary qualitative work shows that people with intellectual disabilities and clinicians are inclined to introduce digital interventions into the routine treatment of mental disorders [27]. In one of the studies on the treatment of anxiety and depression among people with mild to moderate intellectual disabilities [28], the Pesky Gnats program: THE FEEL GOOD ISLAND [29] was used. This study was the first formal evaluation of the efficacy of computerized cognitive behavioral therapy in people with intellectual disabilities and proved that digital mental health interventions can be used successfully in this group of patients. Due to the fact that intellectually disabled patients constitute a very diverse group with different needs, it may be a good idea to use hybrid therapy, i.e., some meetings take place face-to-face with a doctor and some online with the use of electronic devices [30].

Another aspect of helping people with intellectual disabilities is preventing dangerous behavior, which is a physical threat to caregivers and people from the environment, often caused by a deficit in verbal communication [31]. In order to provide support in this field, a number of digital interventions can be used, i.e., mobile applications that could facilitate communication and prevent dangerous behaviors, real-time behavior monitoring applications, helpful in the analysis of behavior patterns, and the social media used by caregivers of people with intellectual disability in order to contact people with similar experiences and obtain support [3].

It should be emphasized that people with intellectual disabilities often also suffer from comorbidities such as diabetes, kidney diseases, respiratory diseases, obesity, etc. The use of telemedicine may be a good solution for them as a way of dealing with their other health problems. Research shows that children with intellectual disabilities have difficulties in acquiring the skills to deal with health problems and gaining access to necessary health care [32]. As a result, it exposes them to the risk of failure to obtain higher education, get a job or a chance to live independently. Moreover, the results of the study by Reichard et al. (2011) proved that people with intellectual disabilities are characterized by worse health and receive fewer preventive benefits than non-disabled people with the same health conditions [33]. Rimmer et al. (2010), studying adolescents with autism and Down’s syndrome, concluded that this group is two to three times more likely to be obese than adolescents in the general population. Moreover, obese youth with intellectual disabilities have secondary health problems, i.e., arterial hypertension, dyslipidemia, diabetes, depression, fatigue, and low self-esteem [34].

The introduction of effective methods of treating chronic diseases, i.e., diabetes, kidney diseases, allergies, asthma, anemia and obesity in people with intellectual disabilities, is extremely important due to their progredient nature and the worsening of health problems as the aging process progresses. Therefore, it is very important that people with intellectual disabilities learn the skills necessary for self-control throughout their lives [35,36,37]. With the advent of modern technologies, people with intellectual disabilities have a chance to learn to meet health needs in a more accessible format and understand the relationship between their knowledge about health needs and their activities in the form of healthy lifestyle and physical activity [38]. A literature review of 15 studies on the use of iPods, iPads and iPhones in learning, communication and learning for people with intellectual disabilities showed positive results in the use of intelligent technologies to learn targeted skills in areas of somatic conditions [39]. Intelligent technology, tailored to the needs of a patient with intellectual disability, suffering from chronic diseases, allows individually adjustments to be made in the control to their health condition, e.g., in a patient with diabetes—strict monitoring of glycemia or the number of calories consumed is essential to reduce the potential damage to organs associated with this disease. Glucose fluctuations can be monitored using the tele-sensor and the app on a smart device, allowing the patient to react and prevent episodes of hypo- or hyper-glycemia. Obviously, the real-time glycemic control helps to reduce the risk of microvascular and macrovascular complications [40]. 

Additionally, with use of modern technology, patients can be monitored in their home environments simply by providing information collected at home and transmitted via smart technology. Information about the state of health, i.e., blood pressure, body weight, and blood glucose levels can be interpreted by the healthcare team, which makes recommendations for the patient without the need to visit the doctor’s office [41]. An effective method in teaching people with intellectual disabilities to use electronic devices may be video modeling. It is an instructional method in which someone shows how to perform certain activities from beginning to end in video format [42]. It has been shown that this is an effective activation tool for people with intellectual disabilities [43,44,45].

During the COVID-19 pandemic, telemedicine appears also to be a very useful platform for patients with chronic neurological conditions including cerebral palsy, as the decreased access to healthcare and therapies requires organizing online meetings of the specialists, patients and their families. One of the proposed forms of such online support is Tele-Guide-lined and Structured Continuous Care (TGSCC), which is a multidisciplinary form of meetings, in which multiple specialists are assembled [46]. Another program, which is addressed to autistic children is Autism Speaks, which provides useful information and resources for children and adults affected with Autism spectrum disorders. Some of the information includes practical tips for parents, useful websites school services, and information [47,48]. During the current pandemic, website-based information with pictograms, ready for download, can be a method to help people with mental intellectual disabilities and their caregivers to learn and rehearse techniques to keep themselves and others safe by using adequate hygiene precautions [49].

Table 2 summarizes the various telemedicine support options for medical care for people with intellectual disabilities.

## 7. Risks Related to the Use of Telemedicine in People with Intellectual Disabilities

The use of digital technology among people with intellectual disabilities brings certain dangers, which in this group of people may be greater than in the general population. People with intellectual disabilities are more prone to disinformation when online. Health information on websites is frequently not standardized and may be of poor quality or false. Moreover, such persons are more susceptible to victimization on social media and are more likely to be victims of cyberbullying due to communication impairment and social isolation [50]. Another problem is that some applications use sensitive data and use it for commercial purposes. Users with intellectual disabilities may have difficulties understanding the privacy policy of some applications, and may be prone to abuse [51]. 

## 8. Limitations of Studies on the Use of Telemedicine in People with Intellectual Disabilities

We also need to be aware of the limitations of the accessible studies on the use of the telemedicine tools for patients with intellectual disability. There is still a paucity of studies focusing on the use of technology to support and promote the prevention, diagnosis, treatment, and monitoring of diseases and the management of health conditions, tailored to individuals with intellectual disability. These studies present several limitations: terms like “telemedicine”, “telehealth” and “e-health” are often used interchangeably for a broad range of support services and technologies, different aims and interventions, small sample size and different type of selection procedure, different age range of patients, poor protocols for a randomized controlled trial, compliance level of parents or caregivers, different use of the assistive products, different internet broadband services. It is therefore difficult to draw firm conclusions about the benefits and limitations of such a modality in people with intellectual disabilities [52,53].

## 9. The New Vision of the Application for People with Intellectual Disabilities

According to the authors, the key to telemedicine care for people with intellectual disabilities is the creation of multi-modal telemedicine applications adapted to the health needs and capabilities of these people. The authors are currently working on a project to create such a modern telemedicine application that can bring significant progress in the care of people with intellectual disabilities. The previous experience, literature search and clinical observations from caring for this group of patients indicate that such a model telemedicine system, in addition to the existing solutions, should also include the following modules:-Pharmacological treatment with feedback function, given by the caregiver; information correlating drug use with their effectiveness can be collected in a database.-Behavioral interventions programmed by a behavioral psychotherapist, the effectiveness of which can, as in the case of pharmacological interventions, be assessed and introduced into the system by the caregiver.-Video interviews, enabling remote contact of the therapist with the patient and his/her caregiver.-A voice analysis module, recording the patient’s voice and providing feedback for the system to correlate it with the effectiveness of pharmacological and behavioral interventions.-A video recording module that allows for the analysis of the patient’s kinetics and facial expressions, enabling the caregiver to record the behavior of a person, which can be analyzed by the telemedicine system, providing the system with further medical data.-Cognitive training, which may be based on simple mobile or computer games, which may, on the one hand, support cognition and, on the other hand, provide relaxation and be an effective strategy for dealing with negative emotions and behaviors.

We are currently going to start a multi-center study of such a telemedicine system for the treatment of people with intellectual disabilities. After confirming its clinical operation, the system can be implemented as a supporting tool for the treatment of intellectually disabled people. 

The scheme of the multimodule telemedicine system, including various modules integrated in one system for mentally disabled persons is shown in Figure 1. 

## 10. Conclusions

Telemedicine can be an effective method of supporting the treatment and rehabilitation of intellectually disabled patients; in particular, when access to traditional healthcare is limited, such as during the COVID-19 pandemic or in rural areas. The use of modern mobile technologies creates new opportunities to improve the quality and accessibility of medical care for a larger number of people with intellectual disabilities. Although there are several barriers for these people in their access to modern technology, more and more telemedicine applications and electronic devices have been designed to match their needs. There is an increasing number of examples of such applications, as well as the tele-visits programs, effectiveness of which was confirmed in clinical studies. They can be supportive not only in the psychological development and rehabilitation, but also in the treatment of general medical conditions like diabetes, kidney diseases, respiratory diseases, obesity and neurological disorders. Multimode mobile applications, tailored to the people with mental disabilities, and using AI, may revolutionize and improve the level of medical assistance for them [54]. In addition to AI, robotics and the combination of AI and robotics may be useful in the therapy of patients with intellectual disabilities [55]. 

## Figures and Tables

**Figure 1 ijerph-18-01746-f001:**
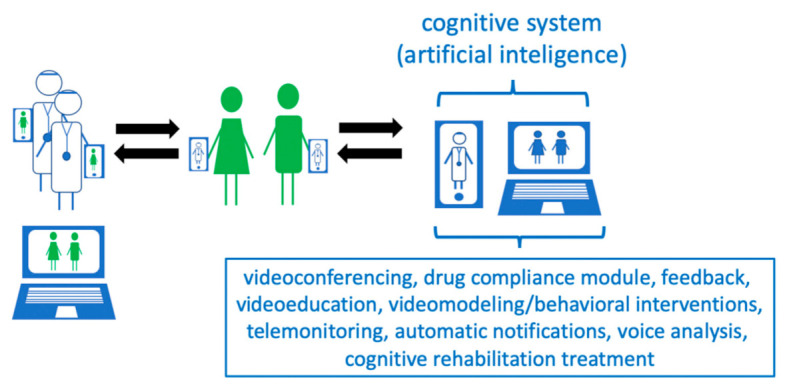
Project of a telemedical care system for people with intellectual disabilities.

**Table 1 ijerph-18-01746-t001:** Results of search results for original studies on the use of telemedicine in the therapy of patients with intellectual disabilities.

Authors	Year	n	Characteristics of Participants	Intervention Used	Results	Conclusions
Salgado T. et al. [18]	2018	52	Patients with developmental disabilities and their caregivers as well as specialists in the field of intellectual disability.	Based on the available functions in telemedicine applications (42 functions grouped into 4 modules (drug list, drug reminder, drug administration record and additional functions)), the respondents were to come to a consensus as to the most preferred ones for controlling drug intake. Consensus for the first, second and third rounds was defined as ≥90%, ≥80%, and ≥75% agreement, respectively.	In addition to the drug list, medication reminders, and drug administration recording functions, it was shown that experts selected three of the most important additional functions: automatic drug refilling in pharmacies; the ability to share information about drugs from the application with suppliers; and the ability to share information about drugs from the application with family, friends and caregivers.	Telemedicine applications containing the indicated functions may be effective in the care of patients with intellectual disabilities.
Ptomey L. et al. [19]	2017	31	An intervention study involving adolescents with developmental and intellectual disabilities.	The subjects participated in 30 min group sessions of physical activity 3 times a week. Classes were held at their homes through videoconferences on a tablet computer for 12 weeks.	Thirty-one patients enrolled and 29 completed the 12-week intervention. Participants participated in 77.2% ± 20.8% of scheduled sessions with an average of 26.7 ± 2.8 min PA/session, with 11.8 ± 4.8 min at moderate to high intensity.	The use of telemedicine techniques may enable increased physical activity in patients with developmental and intellectual disabilities.
Taber-Doughty T. et al. [20]	2010	4	Clinical trial in patients with intellectual disability.	Traditional care and teleconsultation took place alternately for 6 weeks. The subjects were to perform specific tasks, such as addressing the envelope, cooking pudding, brewing tea, etc. Teleconsultation was conducted using the Voice over Internet Protocol (VoIP) protocol, which provided audio communication over the Internet.	It has been shown that the use of tele-visits, despite the fact that it extended the time of task completion, increased the independence of the respondents.	The use of telemedicine techniques may increase the independence of patients with intellectual disabilities in performing daily activities.

**Table 2 ijerph-18-01746-t002:** The use of telemedicine among people with intellectual disabilities and in patients with specific conditions.

Proved telemedicine functionality in care of people with mental disability; communication with the doctor via electronic devices
**Proved Telemedicine Functionality in Care of People with Intellectual Disability**
communication with the doctor via electronic devices health information websites specially adapted to the needs of people with intellectual disabilities video modeling providing people with intellectual disabilities with accurate instructions on how to use electronic devices automatic sending of health information to the doctor applications reminding people with intellectual disabilities about visits, drug intake applications with health monitoring function
**Proved telemedicine functionality in care of people with specific conditions**
multidisciplinary forms of meetings for patients with cerebral palsy, their families and specialists treating them programs addressed to autistic children providing useful information and resources for children and adults affected with Autism Spectrum Disorders

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
