# Peer review of "Telemedicine Treatment and Care for Patients with Intellectual Disability"

_ijerph, 2021, doi:10.3390/ijerph18041746_

Round 1

Reviewer 1 Report

In this work, authors claim to  present a review and  the potential threats related to the use of telemedicine solutions for people with intellectual disabilities. Also, authors introduce a future project for creating a telemedicine system por patients with intellectual disabilities.The research topic is interesting and within the scope of the journal, but this article is not scientifically sound, i.e. does not follow IMRAD.

To improve the article, I suggests the following changes:

Line 2, the title “System of telemedicine treatment and care for intellectually disabled patients” should be changed to reflect the content of the article, which is mostly a review not the system as such. Also I recommend refer to the patiens as "patients with intellectual disabilities" since the condition does not define the person.

Line 74, aim of the authors was to review the existing research. Nevertheless, the information about research databases, research keywords and strings, timeframe of the review, number of articles identified, etc. is missing.

Line 127, it says: MY HEALTH GUIDE" was created, but is should say: MY HEALTH GUIDE was created.

Line 249, Table 1 should be moved to the next page.

Table 1, Bullets should be justified to the left.

Linde 259, the section title states “risk related” but the abstract refers to “potential threats”. I suggest to review and standardize the terminology used along the article.

Line 296, Conclusions title should be in bold and bigger font size.

Line 303, a paragraph should be added explaining the future work planned by the authors.

Author Response

Response to the Reviewer 1

We thank the Reviewer for the interest in our results and for giving us a positive evaluation of the manuscript. We tried to take advantage of the received comments and to improve the manuscript.

In this work, authors claim to  present a review and  the potential threats related to the use of telemedicine solutions for people with intellectual disabilities. Also, authors introduce a future project for creating a telemedicine system por patients with intellectual disabilities.The research topic is interesting and within the scope of the journal, but this article is not scientifically sound, i.e. does not follow IMRAD.

To improve the article, I suggests the following changes:

- Line 2, the title “System of telemedicine treatment and care for intellectually disabled patients” should be changed to reflect the content of the article, which is mostly a review not the system as such. Also I recommend refer to the patiens as "patients with intellectual disabilities" since the condition does not define the person.

The title of the manuscript was changed:

Telemedicine treatment and care for patients with intellectual disability

- Line 74, aim of the authors was to review the existing research. Nevertheless, the information about research databases, research keywords and strings, timeframe of the review, number of articles identified, etc. is missing.

We have added the missing information about the methodology and search results:

  1. Methods

                This review was focused on the adoption on telemedicine tools in the teatment of patients with intellectual disabilities. In order to achieve this result the following data-bases PubMed, Web of Science and Google Scholar were searched (effective date 20.12.2020). The search was performed according to the PICO framework (P - patient, problem or popluation, I - intervention, C- comparison, control or comparator, O - out-comes. During our search we used following terms: intellectual disability (Title/Abstract), telemedicine, telecare (Title/Abstract), traditional consultations (Title/Abstract), psychical, physical activity improvement (Title/Abstract).

  1. Telemedicine applications for people with intellectual disabilities

                The search provided with the use of above terms resulted in finding 16 manuscripts, however after limiting the selection only to original papers fulfilling the criteria of good quality research studies, the papers listed in the table (Table 1) were included into the analysis:

- Line 127, it says: MY HEALTH GUIDE" was created, but is should say: MY HEALTH GUIDE was created.

We have corrected to MY HEALTH GUIDE

- Line 249, Table 1 should be moved to the next page.

Table 1 (now named Table 2) was moved to the next page

- Table 1, Bullets should be justified to the left.

Bullets in the Table were justified to the left

- Line 259, the section title states “risk related” but the abstract refers to “potential threats”. I suggest to review and standardize the terminology used along the article.

The terminology in the article was reviewed and standardised.

- Line 296, Conclusions title should be in bold and bigger font size.

Conclusions are now in bold and bigger font size

- Line 303, a paragraph should be added explaining the future work planned by the authors.

A paragraph was added on the planned future work:

We are currently going to start a multi-center study of such the telemedicine system for the treatment of people with intellectual disabilities. After confirming its clinical operation, the system can be implemented as a supporting tool for the treatment of intellectually disabled people.

Reviewer 2 Report

The manuscript entitled “System of telemedicine treatment and care for intellectually disabled patients” addresses an important field of study, that is to explore possible applications of telemedicine solutions to provide or support services to people with intellectual disability.

This topic has been of great interest over the last twenty years, see for example the studies by Wheeler (1998), Karp et al. (2000), Harper (2006), Buono &. Città (2007).

There is still paucity of studies focusing on the use of technology to support and promote the prevention, diagnosis, treatment, and monitoring of diseases, in a word, the management of health conditions, tailored to individuals with intellectual disability. These studies present several limitations: terms like “telemedicine”, “telehealth” and “e-health” are often used interchangeably for a broad range of support services and technologies, different aims and interventions, small sample size and different type of selection procedure, different age range of patients, poor protocols for a randomized controlled trial; compliance level of parents or caregivers, different use of the assistive products, different internet broadband services. It is therefore difficult to draw firm conclusions about the benefits and limitations of such a modality in people with intellectual disabilities (see Vázquez et al., 2018; Madhavan, 2019).

The point of strength of the manuscript is represented by information regarding a project aimed to devise a modern telemedicine application, the “multimodule telemedicine system”, that can bring about significant advances in the care of people with intellectual disabilities. I also congratulate with the Authors for all their efforts and the time spent on this project.

I would recommend this paper for publication with the following minor revision.

In the “aims” section of the study, the authors review the existing research in the literature to explore possible applications of telemedicine solutions in the care of people with intellectual disabilities, indicating the possibilities, the limitations, and the potential risks associated with its use.

I would recommend that the authors add a brief section on the limitations of telemedicine solutions in the care of people with intellectual disabilities.

References

Buono S., Città S. (2007). Tele-assistance in intellectual disability. Journal of Telemedicine and Telecare. 13(5):241-245.

Dennis C. Harper (2006) Telemedicine for Children With Disabilities, Children's Health Care, 35:1, 11-27.

Karp, W. B., Grigsby, R. K., McSiggan-Hardin, M., Pursley-Crotteau, S., Adams, L. N., Bell, W., et al. (2000). Use of telemedicine for children with special health care needs. Pediatrics, 105, 843–847.

Madhavan, G. (2019). Telepsychiatry in intellectual disability psychiatry: Literature review. BJPsych Bulletin, 43(4), 167-173. doi:10.1192/bjb.2019.5).

Vázquez A, Jenaro C, Flores N, Bagnato MJ, Pérez MC and Cruz M (2018) E-Health Interventions for Adult and Aging Population With Intellectual Disability: A Review. Front. Psychol. 9:2323. doi: 10.3389/fpsyg.2018.02323.

Wheeler, T. (1998). Telemedicine and special needs children. Telehealth Today, 6, 16–20.

Author Response

Response to the Reviewer 2

We thank the Reviewer for the interest in our results and for giving us a positive evaluation of the manuscript. We tried to take advantage of the received comments and to improve the manuscript.

The manuscript entitled “System of telemedicine treatment and care for intellectually disabled patients” addresses an important field of study, that is to explore possible applications of telemedicine solutions to provide or support services to people with intellectual disability.

 This topic has been of great interest over the last twenty years, see for example the studies by Wheeler (1998), Karp et al. (2000), Harper (2006), Buono &. Città (2007).

There is still paucity of studies focusing on the use of technology to support and promote the prevention, diagnosis, treatment, and monitoring of diseases, in a word, the management of health conditions, tailored to individuals with intellectual disability. These studies present several limitations: terms like “telemedicine”, “telehealth” and “e-health” are often used interchangeably for a broad range of support services and technologies, different aims and interventions, small sample size and different type of selection procedure, different age range of patients, poor protocols for a randomized controlled trial; compliance level of parents or caregivers, different use of the assistive products, different internet broadband services. It is therefore difficult to draw firm conclusions about the benefits and limitations of such a modality in people with intellectual disabilities (see Vázquez et al., 2018; Madhavan, 2019).

The point of strength of the manuscript is represented by information regarding a project aimed to devise a modern telemedicine application, the “multimodule telemedicine system”, that can bring about significant advances in the care of people with intellectual disabilities. I also congratulate with the Authors for all their efforts and the time spent on this project.

I would recommend this paper for publication with the following minor revision.

In the “aims” section of the study, the authors review the existing research in the literature to explore possible applications of telemedicine solutions in the care of people with intellectual disabilities, indicating the possibilities, the limitations, and the potential risks associated with its use.

I would recommend that the authors add a brief section on the limitations of telemedicine solutions in the care of people with intellectual disabilities.

# A Section on the limitations of telemedicine solutions was added:

  1. "Limitations of studies on the use of telemedicine in people with intellectual disabilities

We also need to be aware of the limitations of the accessible studies on the use of the telemedicine tools for patients with intellectual disability. There is still paucity of studies focusing on the use of technology to support and promote the prevention, diagnosis, treatment, and monitoring of diseases and the management of health conditions, tailored to individuals with intellectual disability. These studies present several limitations: terms like “telemedicine”, “telehealth” and “e-health” are often used interchangeably for a broad range of support services and technologies, different aims and interventions, small sample size and different type of selection procedure, different age range of patients, poor protocols for a randomized controlled trial; compliance level of parents or caregivers, different use of the assistive products, different internet broadband services. It is therefore difficult to draw firm conclusions about the benefits and limitations of such a modality in people with intellectual disabilities [52, 53]."

 References

Buono S., Città S. (2007). Tele-assistance in intellectual disability. Journal of Telemedicine and Telecare. 13(5):241-245.

Dennis C. Harper (2006) Telemedicine for Children With Disabilities, Children's Health Care, 35:1, 11-27.

Karp, W. B., Grigsby, R. K., McSiggan-Hardin, M., Pursley-Crotteau, S., Adams, L. N., Bell, W., et al. (2000). Use of telemedicine for children with special health care needs. Pediatrics, 105, 843–847.

Madhavan, G. (2019). Telepsychiatry in intellectual disability psychiatry: Literature review. BJPsych Bulletin, 43(4), 167-173. doi:10.1192/bjb.2019.5).

Vázquez A, Jenaro C, Flores N, Bagnato MJ, Pérez MC and Cruz M (2018) E-Health Interventions for Adult and Aging Population With Intellectual Disability: A Review. Front. Psychol. 9:2323. doi: 10.3389/fpsyg.2018.02323.

Wheeler, T. (1998). Telemedicine and special needs children. Telehealth Today, 6, 16–20.

Reviewer 3 Report

This review by Krysta et al presents an overview of telemedicine available to people with intellectual disabilities. However, this is an interesting review, the content lacks critical analysis and authors should consider revising it significantly.

  1. The methodology of the scoping review is not clear? Please elaborate on the methods employed - PICO framework?
  2. The article in its current form appears to be presented with specific articles discussed without critical analyses. It is advised that authors provide a summary table on the list of key studies on the application of telemedicine in patients with intellectual disabilities - effectiveness, challenges etc.
  3. In the section "Telemedicine among people with intellectual disabilities during the COVID-19", authors should add discussion on geographic disparities in telemedicine (see https://pubmed.ncbi.nlm.nih.gov/33178656/). Moreover, the impact on various specialised services available to patients with intellectual disabilities should also be briefly discussed (https://pubmed.ncbi.nlm.nih.gov/33178656/).
  4. In the section, "Experiences of the Berlin Treatment Center for Health", could the authors provide data on a total number of video visits availed by people with intellectual disabilities vs those availing the in-person visits. Has the authors noticed a decline or increase during the COVID-19 pandemic relative to pre-pandemic period?
  5. In the section "The use of telemedicine among people with intellectual disabilities in different health conditions", some discussion on patients with chronic neurological conditions including cerebral palsy should be addressed especially in the context of COVID-19 (https://pubmed.ncbi.nlm.nih.gov/32670193/, https://pubmed.ncbi.nlm.nih.gov/32695066/)
  6. Please revise Table 1 with additional columns on specific conditions e.g., cerebral palsy.
  7. Please define "multimodal telemedicine intervention".
  8. Lines 302-303: "Multimode mobile applications, tailored to the people with mental disabilities, and using AI may revolutionize and improve the level of medical assistance for them [48]." In addition to AI, robotics and combination of AI and robotics maybe useful to patients with intellectual disabilities (see https://pubmed.ncbi.nlm.nih.gov/33224912/)
  9. Please reorganise the last section (297-303) under a separate header "Conclusion". Also, please summarise key findings from the individual sections discussed in the paper.

Author Response

Response to the Reviewer 3

We thank the Reviewer for the interest in our results and for giving us a positive evaluation of the manuscript. We tried to take advantage of the received comments and to improve the manuscript.

This review by Krysta et al presents an overview of telemedicine available to people with intellectual disabilities. However, this is an interesting review, the content lacks critical analysis and authors should consider revising it significantly.

  1. The methodology of the scoping review is not clear? Please elaborate on the methods employed - PICO framework?

We have added a new Section "Mathods" describing the methods employed.PICO framework was applied.

Methods

                This review was focused on the adoption on telemedicine tools in the teatment of patients with intellectual disabilities. In order to achieve this result the following data-bases PubMed, Web of Science and Google Scholar were searched (effective date 20.12.2020). The search was performed according to the PICO framework (P - patient, problem or popluation, I - intervention, C- comparison, control or comparator, O - out-comes. During our search we used following terms: intellectual disability (Title/Abstract), telemedicine, telecare (Title/Abstract), traditional consultations (Title/Abstract), psychical, physical activity improvement (Title/Abstract).

  1. The article in its current form appears to be presented with specific articles discussed without critical analyses. It is advised that authors provide a summary table on the list of key studies on the application of telemedicine in patients with intellectual disabilities - effectiveness, challenges etc.

A summary table was provided -Table 1.

  1. In the section "Telemedicine among people with intellectual disabilities during the COVID-19", authors should add discussion on geographic disparities in telemedicine (see https://pubmed.ncbi.nlm.nih.gov/33178656/). Moreover, the impact on various specialised services available to patients with intellectual disabilities should also be briefly discussed (https://pubmed.ncbi.nlm.nih.gov/33178656/).

A Section on gegraphical disparities was added:

For some countries the fast implementation of telemedicine tools may be complicated by geographic disparities. For example the US is the frontrunner in the access tele-health, and on the other hand Bangladesh or Caribbean countries are in the entry stage of its adoption. Other countries like China, Spain, Italy, UK, Australia, Latin American countries are on varying levels of access to this type of support, with several problems like regional differences, technological and legal aspects still to be solved [23].

  1. In the section, "Experiences of the Berlin Treatment Center for Health", could the authors provide data on a total number of video visits availed by people with intellectual disabilities vs those availing the in-person visits. Has the authors noticed a decline or increase during the COVID-19 pandemic relative to pre-pandemic period?

Unfortunately, due to a lack of structured documentation systems at this hospitals there are no quantitative data.

  1. In the section "The use of telemedicine among people with intellectual disabilities in different health conditions", some discussion on patients with chronic neurological conditions including cerebral palsy should be addressed especially in the context of COVID-19 (https://pubmed.ncbi.nlm.nih.gov/32670193/, https://pubmed.ncbi.nlm.nih.gov/32695066/)

A section on the use of telemedicine in chronic neurological conditions was added:

During the COVID-19 pandemic telemedicine appears also to be a very useful plat-form for patients with chronic neurological conditions including cerebral palsy, as the decreased access to healthcare and therapies requires organizing online meetings of the specialists, patients and their families. One of the proposed form of such online support is Tele-Guide-lined and Structured Continuous Care (TGSCC), which is a multidisciplinary form of meetings, in which multiple specialists are assembled [46]. Another program, which is addressed to autistic children is Autism Speaks, which  provides useful infor-mation and resources for children and adults affected with Autism spectrum disorders. Some of the information includes practical tips for parents, useful websites school ser-vices, and information [47, 48].

  1. Please revise Table 1 with additional columns on specific conditions e.g., cerebral palsy.

In the Table 1 (now named Table 2) additional information on specific conditions was addedd

  1. Please define "multimodal telemedicine intervention".

The required definition can be found now in the following sentence:

The scheme of the multimodule telemedicine system, including various modules inte-grated in one system for mental disabled persons is shown on the Figure 1.

  1. Lines 302-303: "Multimode mobile applications, tailored to the people with mental disabilities, and using AI may revolutionize and improve the level of medical assistance for them [48]." In addition to AI, robotics and combination of AI and robotics maybe useful to patients with intellectual disabilities (see https://pubmed.ncbi.nlm.nih.gov/33224912/)

The sentence referring to AI and robotics was added:

In addition to AI, robotics and combination of AI and robotics may be useful in the therapy of patients with intellectual disabilities [55].

  1. Please reorganise the last section (297-303) under a separate header "Conclusion". Also, please summarise key findings from the individual sections discussed in the paper.

The Conclusion Section was reorganised with a summary of the key findings from individual sections:

Telemedicine can be an effective method of supporting the treatment and rehabili-tation of intellectually disabled patients. In particular, when access to traditional healthcare is limited, such as during the COVID-19 pandemic or in rural areas. The use of modern mobile technologies creates new opportunities to improve the quality and accessibility of medical care for a larger number of people with intellectual disabilities. Although there are several barriers for these people in their access to modern technology, more and more telemedicine applications and electronic devices have been designed to match their needs. There is an increasing number of examples of such applications, as well as the tele-visits programs, effectiveness of which was confirmed in clinical studies. They can be supportive not only in the their psychological development and rehabilitation, but also in the treatment in general medical conditions like diabetes, kidney diseases, respiratory diseases, obesity and neurological disorders. Multimode mobile applications, tailored to the people with mental disabilities, and using AI may revolutionize and im-prove the level of medical assistance for them [54]. In addition to AI, robotics and combination of AI and robotics may be useful in the therapy of patients with intellectual disabilities [55].

Round 2

Reviewer 1 Report

The manuscript has improved significantly and I believe is ready to be published. Congrats!

Minor suggestions:

  • Line 33. It says "we used following terms:", it should say "we used the following terms:"
  • Line 90. It says "in the table (Table 1) were included", it should say "in Table 1 were included"
  • Line 343. It says "treatment of intellectually disabled people.", it should say "treatment of people with intellectual disabilities."

Reviewer 3 Report

The authors have addressed the concerns raised.